# Speaker Diarization as a Fully Online Learning Problem in MiniVox

**Baihan Lin** [1]   **Xinxin Zhang** [2]

## Abstract

We proposed a novel AI framework to conduct real-time multi-speaker diarization and recognition without prior registration and pretraining in a fully online learning setting. Our contributions are two-fold. First, we proposed a new benchmark to evaluate the rarely studied fully online speaker diarization problem. We built upon existing datasets of real world utterances to automatically curate *MiniVox*, an experimental environment which generates infinite configurations of continuous multi-speaker speech stream. Secondly, we considered the practical problem of online learning with episodically revealed rewards and introduced a solution based on semi-supervised and self-supervised learning methods. Lastly, we provided a workable web-based recognition system which interactively handles the cold start problem of new user's addition by transferring representations of old arms to new ones with an extendable contextual bandit. We demonstrated that our proposed method obtained robust performance in the online MiniVox framework. [1]

## 1. Introduction

Speaker recognition involves two essential steps: registration and identification (Tirumala et al., 2017). In laboratory setting, the state-of-the-art approaches usually emphasize the registration step with deep networks (Snyder et al., 2018) trained on large-scale speaker profile dataset (Nagrani et al., 2017). However, in real life, requiring all users to complete voiceprint registration before a multi-speaker teleconference is hardly a preferable way of system deployment. Dealing with this challenge, speaker diarization is the task to partition an audio stream into homogeneous segments according to the speaker identity (Anguera et al., 2012). Recent advancements have enabled (1) contrastive audio embedding extractions such as Mel Frequency Cepstral Coefficients (MFCC) (Hasan et al., 2004), i-vectors (Shum et al., 2013) and d-vectors (Wang et al., 2018); (2) effective clustering modules such as Gaussian mixture models (GMM) (Zajíc et al., 2017), mean shift (Senoussaoui et al., 2013), Kmeans and spectral clustering (Wang et al., 2018) and supervised Bayesian non-parametric methods (Fox et al., 2011; Zhang et al., 2019); and (3) reasonable resegmentation modules such as Viterbi and factor analysis subspace (Sell & Garcia-Romero, 2015). In this work, we proposed a new paradigm to consider the speaker diarization as a fully online learning problem of the speaker recognition task: it combines the embedding extraction, clustering and resegmentation into the same problem as an online decision making problem.

**Why is this online learning problem different?** The state-of-the-art speaker diarization systems usually require large datasets to train their audio extraction embeddings and clustering modules, especially the ones with deep neural networks and Bayesian nonparametric models. In many real-world applications in developing countries, however, the training set can be limited and hard to collect. Since these modules are pretrained, applying them to out-of-distribution environments can be problematic. For instance, an intelligent system trained with American elder speaker data might find it hard to generalize to a Japanese children diarization task because both the acoustic and contrastive features are different. To tackle this problem, we want the system to learn continually. To push this problem to the extreme, we are interested in a fully online learning setting, where not only the examples are available one by one, the agent receives no pretraining from any training set before deployment, and learns to detect speaker identity on the fly through reward feedbacks. To the best of our knowledge, this work is the first to consider diarization as a fully online learning problem. Through this work, we aim to understand the extent to which diarization can be solved as merely an online learning problem and whether traditional online learning algorithms (e.g. contextual bandits) can be beneficial to provide a practical solution.

[1]Department of Applied Mathematics, University of Washington, Seattle, WA 98195, USA [2]Department of Electrical and Computer Engineering, University of Washington, Seattle, WA 98195, USA. Correspondence to: Baihan Lin <baihan.lin@columbia.edu>.

*The 37$^{th}$ International Conference on Machine Learning*, Vienna, Austria, PMLR 119, 2020. Copyright 2020 by the author(s).

[1]The web-based application of a real-time system can be accessed at https://www.baihan.nyc/viz/VoiceID/. The code for benchmark evaluation can be accessed at https://github.com/doerlbh/MiniVox

**What is a preferable online speaker diarization system?**
A preferable AI engine for such a realistic speaker recognition and diarization system should (1) not require user registrations, (2) allow new user to be registered into the system real-time, (3) transfer voiceprint information from old users to new ones, (4) be up running without pretraining on large amount of data in advance. While attractive, assumption (4) introduced an additional caveat that the labeling of the user profiles happens purely on the fly, trading off models pretrained on big data with the user directly interacting with the system by correcting the agent as labels. To tackle these challenges, we formulated this problem into an interactive learning model with cold-start arms and episodically revealed rewards (users can either reveal no feedback, approving by not intervening, or correcting the agent).

**Why do we need a new benchmark?** Traditional dataset in the speaker diarization task are limited: CALLHOME American English (Canavan et al., 1997) and NIST RT-03 English CTS (Martin & Przybocki, 2000) contained limited number of utterances recorded under controlled conditions. For online learning experiments, a learn-from-scratch agent usually needs a large length of data stream to reach a comparable result. Large scale speaker recognition dataset like VoxCeleb (Nagrani et al., 2017; 2019) and Speakers in the Wild (SITW) (McLaren et al., 2016) contained thousands of speaker utterances recorded in various challenging multi-speaker acoustic environments, but they are usually only used to pretrain diarization embeddings. In this work, we proposed a new benchmark called *MiniVox*, which can transform any large scale speaker identification dataset into infinitely long audio streams with various configurations.

We built upon LinUCB (Li et al., 2010) and proposed a semi-supervised learning variant to account for the fact that the rewards are entirely missing in many episodes. For each episode without feedbacks, we applied a self-supervision process to assign a pseudo-action upon which the reward mapping is updated. Finally, we generated new arms by transferring learned arm parameters to similar profiles given user feedbacks.

## 2. The Fully Online Learning Problem

Algorithm 1 presents at a high-level our problem setting, where $c(t) \in \mathbb{R}^d$ is a vector describing the context at time

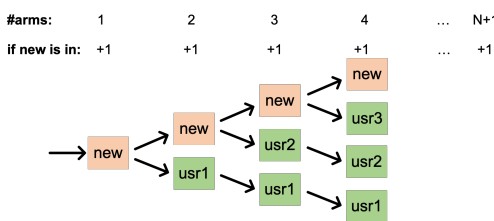

*Figure 1.* Arm expansion process of the bandit agents.

$t$, $r_a(t) \in [0, 1]$ is the reward of action $a$ at time $t$, and $r(t) \in [0, 1]^K$ denotes a vector of rewards for all arms at time $t$. $\mathbb{P}_{c,r}$ denotes a joint probability distribution over $(c, r)$, and $\pi : C \to A$ denotes a policy. Unlike traditional setting, in step 5 we have the rewards revealed in an episodic fashion (i.e. sometimes there are feedbacks of rewards being 0 or 1, sometimes there are no feedbacks of any kind). We consider our setting an online semi-supervised learning problem (Yver, 2009; Ororbia et al., 2015), where the agent learns from both labeled and unlabeled data in online setting.

---

**Algorithm 1** Online Learning with Episodic Rewards

---

1: **for** t = 1,2,3,$\cdots$, T **do**
2:    $(c(t), r(t))$ is drawn according to $\mathbb{P}_{c,r}$
3:    $c(t)$ is revealed to the player
4:    Player chooses an action $i = \pi_t(c(t))$
5:    Feedbacks $r_a(t)$ for all arms are episodically revealed
6:    Player updates its policy $\pi_t$
7: **end for**

---

## 3. Proposed Online Learning Solution

### 3.1. Contextual Bandits with Extendable Arms

In an ideal online learning scenario without oracle, we start with a single arm, and when new labels arrive new arms are then generated accordingly. This problem is loosely modelled by the bandits with infinitely many arms (Berry et al., 1997). For our specific application of speaker registration process, we applied the arm expansion process outlined in Figure 1: starting from a single arm (for the "new" action), if a feedback confirms a new addition, a new arm is initialized and appended to the arm list.

### 3.2. Episodically Rewarded LinUCB

We proposed Background Episodically Rewarded LinUCB (BerlinUCB), a semi-supervised and self-supervised online contextual bandit which updates the context representations and reward mapping separately given the state of the feedbacks being present or missing (Algorithm 2). We assume that (1) when there are feedbacks available, the feedbacks are genuine, assigned by the oracle, and (2) when the feedbacks are missing (not revealed by the background), it is either due to the fact that the action is preferred (no intervention required by the oracle, i.e. with an implied default rewards), or that the oracle didn't have a chance to respond or intervene (i.e. with unknown rewards). Especially in the Step 15, when there is no feedbacks, we assign the context $\mathbf{x}_t$ to a class $a'$ (an action arm) with the self-supervision given the previous labelled context history. Since we don't have the actual label for this context, we only update reward mapping parameter $\mathbf{b}_{a'}$ and leave the covariance matrix $\mathbf{A}_{a'}$ untouched. The additional usage of unlabelled data (or unrevealed feedback) is especially important in our model.

*Figure 2.* (A) The **flowchart** of the Online Learning problem and (B) the *MiniVox* **Benchmark**.

---

**Algorithm 2** BerlinUCB

---

1: **Initialize** $c_t \in \mathbb{R}_+, \mathbf{A}_a \leftarrow \mathbf{I}_d, \mathbf{b}_a \leftarrow \mathbf{0}_{d \times 1} \forall a \in \mathcal{A}_t$
2: **for** t = 1,2,3,$\cdots$, T **do**
3:     Observe features $\mathbf{x}_t \in \mathbb{R}^d$
4:     **for all** $a \in \mathcal{A}_t$ **do**
5:         $\hat{\theta}_a \leftarrow \mathbf{A}_a^{-1} \mathbf{b}_a$
6:         $p_{t,a} \leftarrow \hat{\theta}_a^\top \mathbf{x}_t + c_t \sqrt{\mathbf{x}_t^\top \mathbf{A}_a^{-1} \mathbf{x}_t}$
7:     **end for**
8:     Choose arm $a_t =_{a \in \mathcal{A}_t} p_{t,a}$
9:     **if** the background revealed the feedbacks **then**
10:         Observe feedback $r_{a_t,t}$
11:         $\mathbf{A}_{a_t} \leftarrow \mathbf{A}_{a_t} + \mathbf{x}_t \mathbf{x}_t^\top$
12:         $\mathbf{b}_{a_t} \leftarrow \mathbf{b}_{a_t} + r_{a_t,t} \mathbf{x}_t$
13:     **elif** the background revealed NO feedbacks **then**
14:         **if** use self-supervision feedback
15:             $r' = [a_t == \text{predict}(\mathbf{x}_t)]$ % clustering modules
16:             $\mathbf{b}_{a_t} \leftarrow \mathbf{b}_{a_t} + r' \mathbf{x}_t$
17:         **elif**          % ignore self-supervision signals
18:             $\mathbf{A}_{a_t} \leftarrow \mathbf{A}_{a_t} + \mathbf{x}_t \mathbf{x}_t^\top$
19:         **end if**
20:     **end if**
21: **end for**

---

### 3.3. Self-Supervision and Semi-Supervision Modules

We construct our self-supervision modules given the cluster assumption of the semi-supervision problem: the points within the same cluster are more likely to share a label. As shown in many work in modern speaker diarization, clustering algorithms like GMM (Zajíc et al., 2017), mean shift (Senoussaoui et al., 2013) and spectral clustering (Wang et al., 2018) are especially powerful unsupervised modules, especially in their offline versions. Their online variants, however, often performs poorly (Zhang et al., 2019). As in this work, we focus on the completely online setting, we chose three popular clustering algorithms as self-supervision modules: GMM, Kmeans and K-nearest neighbors.

### 3.4. Complete Engine for Online Speaker Diarization

To adapt our BerlinUCB algorithm to the specific application of speaker recognition, we first define our actions. There are three major classes of actions: an arm "New" to denote that a new speaker is detected, an arm "No Speaker" to denote that no one is speaking, and N different arms "User n" to denote that user n is speaking. Table 1 presents the reward assignment given four types of feedbacks. Note that we assume that when the agent correctly identifies the speaker (or no speaker), the user (as the feedback dispenser) should send no feedbacks to the system by doing nothing. In another word, in an ideal scenario when the agent does a perfect job by correctly identifying the speaker all the time, we are not necessary to be around to correct it anymore (i.e. truly feedback free). As we pointed out earlier, this could be a challenge earlier on, because other than implicitly approving the agent's choice, receiving no feedbacks could also mean the feedbacks are not revealed properly (e.g. the human oracle took a break). Furthermore, we note that when "No Speaker" and "User n" arms are correctly identified, there is no feedback from us the human oracle (meaning that these arms would never have learned from a single positive reward if we don't use the "None" feedback iterations at all!). The semi-supervision by self-supervision step is exactly tailored for a scenario like this, where the lack of revealed positive reward for "No Speaker" and "User n" arms is compensated by the additional training of the reward mapping $\mathbf{b}_{a_t}$ if context $\mathbf{x}_t$ is correctly assigned.

To tackle the cold start problem, the agent grows it arms in the following fashion: the agent starts with two arms, "No Speaker" and "New"; if it is actually a new speaker speaking, we have the following three conditions: (1) if "New" is chosen, the user approves this arm by giving it a positive reward (i.e. clicking on it) and the agent initializes a new arm called "User $N$" and update $N = N + 1$ (where $N$ is the number of registered speakers at the moment); (2)

| Feedback types | (+,+) | (+,-) | (-,+) | None |
|---|---|---|---|---|
| New | $r = 1$ | $r = 0$ | | |
| No Speaker | - | $r = 0$ | $r = 0$ | Alg. 2 Step 13 |
| User n | - | $r = 0$ | $r = 0$ | |

*Table 1.* Routes given either no feedbacks, or a feedback telling the agent that the correct label is $a*$. (+,+) means that the agent guessed it right by choosing the right arm; (+,-) means that the agent chose this arm incorrectly, since the correct one is another arm; (-,+) means that the agent didn't choose this arm, while it turned out to be the correct one. "-" means NA.

if "No Speaker" is chosen, the user disapproves this arm by giving it a zero reward and clicking on the "New" instead), while the agent initializes a new arm; (3) if one of the user arms is chosen (e.g. "User 5" is chosen while in fact a new person is speaking), the agent copies the wrong user arm's parameters to initialize the new arm, since the voiceprint of the mistaken one might be beneficial to initialize the new user profile. In this way, we can transfer what has been learned for a similar context representations to the new arm.

## 4. Benchmark Description: *MiniVox*

MiniVox is an automatic framework to transform any speaker-labelled dataset into continuous speech datastream with episodically revealed label feedbacks. Since our online learning problem setting assumes learning the voiceprints without any previous training data at all, *MiniVox*'s flexibility in length and configuration is especially important. As outlined in Figure 2, *MiniVox* has a straightforward data stream generation pipeline: given a pool of single-speaker-annotated utterances, randomly concatenate multiple pieces with a chosen number of speakers and a desired length. The reward stream is then sparsified with a parameter $p$ as the percentage of time a feedback is revealed.

There are two scenarios that we can evaluate in MiniVox: if we assume there is an oracle, the online learning model is given the fixed number of the speakers in the stream; if we assume there is no oracle, the online learning model will start from zero speaker and then gradually discover and register new speakers for future identification and diarization.

## 5. Empirical Evaluation

### 5.1. Experimental Setup and Metrics

We applied MiniVox on VoxCeleb (Nagrani et al., 2017) to generate three data streams with 5, 10 and 20 speakers to simulate real-world conversations. We extracted two types of features (more details in section 5.2) and evaluated it in two scenarios (with or without oracle). The reward streams are sparsified given a revealing probability of 0.5, 0.1, 0.01 and 0.001. In summary, we evaluated our models in a combinatorial total of 3 speaker numbers $\times$ 4 reward revealing probabilities $\times$ 2 feature types $\times$ 2 test scenarios = 48 online learning environments. The online learning timescale

range from $\sim$12000 to $\sim$60000 timeframes. For notation of a specific MiniVox, in this paper we would denote "MiniVox C5-MFCC-60k" as a MiniVox environment with 5 speakers ranging 60k time frames using MFCC as features.

To evaluate the performance, we reported Diarization Error Rates (DER) in the above MiniVox environments. In addition, as a common metric in online learning literature, we also recorded the cumulative reward: at each frame, if the agent correctly predicts a given speaker, the reward is counted as +1 (no matter if the agent observes the reward).

We compared 9 agents: LinUCB is the contextual bandit with extendable arms proposed in section 3.1. BerlinUCB is the standard contextual bandit model designed for sparse feedbacks without the self-supervision modules. We have four baseline models: Kmeans, KNN (with K=5), GMM and a random agent[2]. To test the effect of self-supervision, we introduced three clustering modules in BerlinUCB (alg 2, Step 15) denoted: B-Kmeans, B-KNN, and B-GMM.

### 5.2. Feature Embeddings: MFCC and CNN

We utilized two feature embeddings for our evaluation: MFCC (Hasan et al., 2004) and a Convolutional Neural Network (CNN) embedding. We utilized the same CNN architecture as the VGG-M (Chatfield et al., 2014) used in VolCeleb evaluation (Nagrani et al., 2017). It takes the spectrogram of an utterance as the input, and generate a feature vector of 1024 in layer fc8 (for more details of this CNN, please refer to table 4 in (Nagrani et al., 2017)).

**Why don't we use more complicated embeddings?** Although more complicated embedding extraction modules such as i-vectors (Shum et al., 2013) or d-vectors (Wang et al., 2018) can improve diarization, they require extensive pretraining on big datasets, which is contradictory to our problem setting and beyond our research scope.

**Why do we still include this CNN?** The CNN model was trained for speaker verification task in VoxCeleb and we are curious about the relationship between a learned representation and our online learning agents. Despite this note, we are most interested in the performance given MFCC features, because we aim to push the system *fully online*, to the limit of not having pretraining of any type before deployment.

### 5.3. Results

Given MFCC features without pretraining, our online learning agent demonstrated a relatively robust performance. As shown in Figure 3(a,b,c,d), in many conditions, the proposed contextual bandits significantly outperformed baselines when revealing probability is very low (p=0.01 or 0.1).

---

[2]In the oracle-free case, the random agent randomly selects from the "new" arm and the registered user arms, suggesting a possibility of going to infinitely (and incorrectly) many profiles.

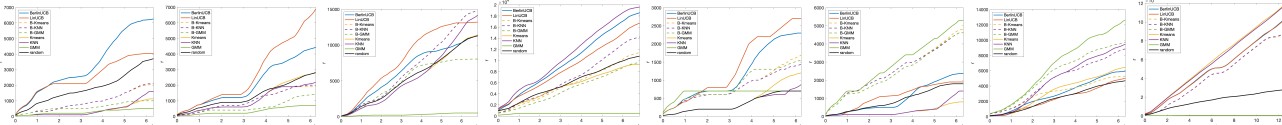

*Figure 3.* **Example reward curve**. Positive: (a) C10-MFCC, p=0.01; (b) C20-MFCC, p=0.01; (c) C5-MFCC, p=0.01; (d) C5-MFCC, p=0.5; (e) C20-MFCC, p=0.01, oracle. Negative: (f) C10-MFCC, p=0.01, oracle; (g) C10-MFCC, p=0.1, oracle; (h) C5-CNN, p=0.5.

*Table 2.* Diarization Error Rate (%) in MiniVox **without** Oracle

|  | MiniVox C5-MFCC-60k | | | MiniVox C5-CNN-12k | | |
|---|---|---|---|---|---|---|
|  | $p = 0.5$ | $p = 0.1$ | $p = 0.01$ | $p = 0.5$ | $p = 0.1$ | $p = 0.01$ |
| BerlinUCB | 71.81 | 80.03 | 82.38 | 17.42 | 32.03 | 65.16 |
| LinUCB | 74.74 | 78.71 | 79.30 | 17.81 | 32.73 | 58.98 |
| B-Kmeans | 82.82 | 79.15 | **77.39** | 28.83 | 63.67 | 82.58 |
| B-KNN | 78.71 | 80.62 | **77.39** | 28.36 | 82.58 | 82.58 |
| B-GMM | 85.32 | 83.41 | 87.67 | 99.61 | 99.61 | 99.69 |
| Kmeans | 86.20 | 85.76 | 82.67 | **5.47** | **8.91** | **40.23** |
| KNN | **70.34** | **72.98** | 78.12 | 6.09 | 13.75 | 53.75 |
| GMM | 99.27 | 99.27 | 99.27 | 99.61 | 99.61 | 99.69 |
| random | 83.41 | 81.50 | 82.97 | 77.89 | 78.98 | 77.66 |
|  | MiniVox C10-MFCC-60k | | | MiniVox C10-CNN-12k | | |
|  | $p = 0.5$ | $p = 0.1$ | $p = 0.01$ | $p = 0.5$ | $p = 0.1$ | $p = 0.01$ |
| BerlinUCB | 82.46 | 85.31 | **89.26** | 42.77 | 57.41 | 74.02 |
| LinUCB | 84.36 | 86.73 | 93.36 | 49.55 | 68.57 | 81.16 |
| B-Kmeans | 91.15 | 92.58 | 96.68 | 60.89 | 70.89 | 99.55 |
| B-KNN | 89.73 | 90.05 | 96.68 | 60.89 | 82.05 | 99.55 |
| B-GMM | 90.21 | 94.63 | 98.42 | 99.20 | 93.57 | 99.64 |
| Kmeans | 92.26 | 94.15 | 98.10 | 10.36 | **18.75** | **47.86** |
| KNN | **79.78** | **84.52** | 97.47 | **9.29** | 31.25 | 70.27 |
| GMM | 98.42 | 98.42 | 99.21 | 99.20 | 99.20 | 99.37 |
| random | 90.21 | 88.78 | 92.89 | 79.29 | 81.34 | 83.75 |
|  | MiniVox C20-MFCC-60k | | | MiniVox C20-CNN-12k | | |
|  | $p = 0.5$ | $p = 0.1$ | $p = 0.01$ | $p = 0.5$ | $p = 0.1$ | $p = 0.01$ |
| BerlinUCB | 88.62 | **87.02** | 92.79 | 41.72 | 59.06 | 83.28 |
| LinUCB | 91.35 | 88.94 | **88.46** | 51.56 | 83.52 | 74.84 |
| B-Kmeans | 95.19 | 95.99 | 96.96 | 72.03 | 75.31 | 99.53 |
| B-KNN | 93.43 | 95.99 | 96.79 | 72.03 | 74.06 | 99.53 |
| B-GMM | 92.79 | 96.31 | 97.76 | 87.73 | 81.09 | 83.28 |
| Kmeans | 90.54 | 93.43 | 95.51 | **6.02** | **12.81** | **54.77** |
| KNN | **86.38** | 89.26 | 95.99 | 8.67 | 32.66 | 75.08 |
| GMM | 96.96 | 97.44 | 98.88 | 98.98 | 98.98 | 99.37 |
| random | 93.59 | 94.07 | 95.35 | 87.03 | 87.73 | 89.69 |

*Table 3.* Diarization Error Rate (%) in MiniVox **with** Oracle

|  | MiniVox C5-MFCC-60k | | | MiniVox C5-CNN-12k | | |
|---|---|---|---|---|---|---|
|  | $p = 0.5$ | $p = 0.1$ | $p = 0.01$ | $p = 0.5$ | $p = 0.1$ | $p = 0.01$ |
| BerlinUCB | 74.89 | 77.24 | 86.93 | 17.27 | 22.19 | 66.02 |
| LinUCB | 72.83 | 78.12 | 76.80 | 17.73 | 32.73 | 58.98 |
| B-Kmeans | 75.33 | 78.27 | 83.11 | 20.55 | 40.70 | 58.98 |
| B-KNN | 77.39 | 77.97 | 83.99 | 20.47 | 41.33 | 58.98 |
| B-GMM | 74.16 | 76.21 | 77.24 | 52.58 | 81.02 | 58.98 |
| Kmeans | 78.41 | 82.82 | 83.11 | **4.06** | **7.42** | **39.53** |
| KNN | 70.63 | 73.27 | 80.47 | 6.64 | 13.75 | 53.52 |
| GMM | **70.34** | **72.54** | **74.74** | 54.38 | 81.02 | 58.98 |
| random | 79.59 | 80.76 | 85.9 | 79.92 | 80.39 | 85.55 |
|  | MiniVox C10-MFCC-60k | | | MiniVox C10-CNN-12k | | |
|  | $p = 0.5$ | $p = 0.1$ | $p = 0.01$ | $p = 0.5$ | $p = 0.1$ | $p = 0.01$ |
| BerlinUCB | 88.31 | 90.21 | 95.89 | 45.18 | 65.27 | 79.38 |
| LinUCB | 84.99 | 91.63 | 97.00 | 50.00 | 72.14 | 65.18 |
| B-Kmeans | 87.84 | 91.47 | 91.94 | 50.27 | 72.50 | 72.32 |
| B-KNN | 86.73 | 85.78 | 92.58 | 49.64 | 72.14 | 77.77 |
| B-GMM | 88.94 | 84.52 | 92.58 | 76.52 | 71.88 | 69.46 |
| Kmeans | 89.42 | 89.57 | 98.74 | 11.16 | **20.27** | **49.49** |
| KNN | 80.25 | 84.68 | 97.79 | **9.55** | 31.25 | 70.45 |
| GMM | 90.36 | **79.62** | **91.63** | 76.52 | 78.30 | 77.77 |
| random | 87.99 | 92.26 | 97.16 | 90.00 | 90.89 | 92.32 |
|  | MiniVox C20-MFCC-60k | | | MiniVox C20-CNN-12k | | |
|  | $p = 0.5$ | $p = 0.1$ | $p = 0.01$ | $p = 0.5$ | $p = 0.1$ | $p = 0.01$ |
| BerlinUCB | 92.31 | 94.55 | 96.31 | 58.75 | 68.98 | 88.83 |
| LinUCB | 89.10 | 93.43 | **95.67** | 53.44 | 70.47 | 83.44 |
| B-Kmeans | 92.95 | 95.67 | 96.96 | 55.16 | 70.86 | 94.06 |
| B-KNN | 91.83 | 92.47 | 97.44 | 54.30 | 89.84 | 96.72 |
| B-GMM | 95.19 | 91.99 | 97.44 | 86.48 | 77.97 | 96.64 |
| Kmeans | 91.67 | 94.23 | 98.08 | **7.66** | **13.75** | **55.63** |
| KNN | **86.86** | **89.26** | 98.08 | 9.690 | 32.73 | 75.08 |
| GMM | 98.08 | 94.87 | 98.88 | 93.52 | 95.08 | 97.11 |
| random | 94.71 | 94.71 | 98.88 | 95.55 | 95.86 | 97.03 |

**Learning without Oracle.** Table 2 reports DER in MiniVox without Oracle. In MFCC environments, we observed that in high-difficulty scenarios (such as C20), the proposed BerlinUCB variants outperformed all the baselines even when the reward revealing probability was as low as 0.01. In low-difficulty scenarios, traditional clustering methods like KNN performed the best, while this benefit was inherited by B-KNN and B-Kmeans when feedbacks were sparse (p=0.01). In the CNN cases, we observed that Kmeans performed the best. This is expected because the CNN model was trained with the constrastive loss for a high verification accuracy (Nagrani et al., 2017). While the clustering modules merely classify the CNN feature by their proximity, our online learning model need to learn about their reward mapping from scratch, while maintaining a good balance between exploitation and exploration.

**Learning with Oracle.** Given the number of speakers, traditional clustering agents performed better (Table 3). However, the behaviors vary: we observed that GMM performed the poorest in the oracle-free environments, but performed the best in the environments with oracle; we also noted that

despite the best model in many oracle-free environments, Kmeans performed poorly in the MFCC environments with oracle. Another winning algorithm, KNN, requires the model to store all historical data points and search through the entire memory, which can be computationally inhibitory in real-world applications. Our online learning models maintains a relatively robust performance by keeping among the top 3 algorithms in most cases with and without oracle.

**Is self-supervision useful?** To our surprise, our benchmark results suggested that the proposed self-supervision modules didn't improve upon both the baseline models and our proposed contextual bandit models. Only in specific conditions (e.g. MiniVox C5-MFCC-60k p=0.01), the self-supervised contextual bandits outperformed both the standard Berlin-UCB and all the baseline. Further investigation into the reward curve revealed more complicated interactions between the self-supervision modules with the online learning modules (the contextual bandit): as shown in Figure 3(f,g,h), B-GMM and B-KNN maintained build upon the effective reward mapping from their BerlinUCB backbone, and benefited from the unlabelled data points to perform fairly well.

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
