# OpenReview forum: "Speaker Diarization as a Fully Online Learning Problem in MiniVox"
_ICML.cc/2020/Workshop/SAS — Submitted to SAS 2020_

### Official Review · AnonReviewer1 · 2020-06-28

**Rating:** 7
**Confidence:** 4

**Review:**

The paper introduces an new online learning framework for speaker diarization as well as  a data framework to data generation. The paper evaluated the framework against several other baselines.

1. The paper is very nicely written, the problem and framework are clearly explained. I very much like the way the results section is written with summaries/ questions to be answered at the beginning of each paragraph.

2. The authors have conducted extensive experiments with various number of speakers, feedback rates, etc. The results are clearly analyzed and explained.

3. The data generating framework by itself is beneficial for the community, it allows the one to generate speech from arbitrary number of speakers, I could see this being useful to study new algorithms/ methods.

4. Though the proposed framework is nice, it seems that self-supervision only helps in limited cases. This result seems rather surprising.

---

### Official Review · AnonReviewer3 · 2020-06-29
**Online speaker diarization: Interesting algorithm, but missing details**

**Rating:** 5
**Confidence:** 4

**Review:**

The introductory Section 1 provides a good overview of the general problem and what challenges have to be met when dealing with speaker diarization in an online processing scenario. It provides suitable references to previous work in the field and introduces available datasets and their respective shortcomings regarding online processing capabilities.

Section 2 provides a brief overview of the online learning problem with episodic rewards, which allows the reader to get the general idea and potential challenges of this task. However, there are some notational inconsistencies when referred to in later sections (see below).

Section 3 can be considered the core part of the paper, where the actual algorithm proposed in the paper is described. However, this section is unfortunately quite brief and it is challenging to understand and grasp the proposed method at first glance. In particular, the following points should be considered here:

Section 3.1: 	The authors mention the bandits with infinitely many arms framework proposed by Berry et al. without describing any details about this method. Some additional background information and a brief summary of the general idea behind this framework would make it easier to follow the new ideas presented in the subsequent sections. Especially regarding Fig. 1 it would have been helpful to provide some additional information about how the growing of arms is handled exactly. Also, there is some notational inconsistency, as the number of arms is denoted as N in Fig.1, whereas it is denoted as K in Sec. 2.

Section 3.2:	The description of the proposed algorithm is lacking some information to fully understand all the details. In Alg. 2, the symbol $p_{t, a}$ is not described in the text. If it is a probability, how can be ensured that it is bounded between 0 and 1? If it is a score that is somewhat proportional to a probability, this should be noted either in the text or in the algorithm description with the additional information that some kind of normalization is required (this is not obvious from its application in line 8). Furthermore,  $c_{t}$ in line 6 of Alg. 2 might be confused with the context variable introduced in Sec. 2, which is denoted as $\boldsymbol{x}_{t}$ here. There is also some additional notation inconsistency, as e.g. the rewards are denoted as $r_{a_{t}, t}$ in Alg. 2, whereas they are denoted as $r_{a}(t)$ in Sec. 2. Also, the function ‘predict’ in line 15 of Alg. 2 is not explained in this section (the hint “clustering methods” in the pseudocode does not provide any meaningful information at this point). The authors should consider some major adjustments in this section during revision to make it more self-contained and provide all required information to fully understand the proposed algorithm.

Section 3.3:	This section should emphasize how GMM, k-Means and KNN are exactly applied in the proposed framework. It is not clear if clustering is performed using the conventional batch-wise approaches or the online clustering variants.

Section 3.4:	The description of how the system is actually implemented provides a good overview on how the framework is supposed to tackle the problem of online speaker diarization. The process of branch management regarding the different bandit arms is described in sufficient detail and the authors provide a good description on how the semi-supervision by self-supervision idea fits into the overall framework. Also, the reward selection scheme applied in this paper makes sense and Tab. 1 provides a good overview on how it is implemented. However, this section misses information on some additional properties of the system, especially regarding potential branch pruning strategies and the processing of very sparse reward feedback. At the end of Sec. 3.4, the process of adding a new user profile is described. In addition, it would have been helpful to at least discuss potential problems that might occur if the number of users grows steadily through misclassifications. One could imagine a scenario with very sparse reward information provided by the oracle, which might lead to the detection of many new users, even though this is actually not the case (e.g. if a single speaker exhibits some variability in speech characteristics). Would it be possible to prune or merge specific arms in this case during online processing to prevent the model from generating a potentially very large number of arms?

Even though section 4 only provides a very short text description, the proposed MiniVox benchmark system is presented very well in Fig. 2.

Section 5 …

Section 5.1:	Diarization Error Rate is a suitable and well-established metric to evaluate the performance of speaker diarization systems. However, it should either be briefly explained here or at least pointed out as a reference to, e.g., “The 2009 (RT-09) rich transcription meeting recognition evaluation plan”, to make it easier to grasp for readers not familiar with the topic. Also, it is not quite clear how the k-Means, KNN and GMM baseline models operate compared to BerlinUCB. Are they also operating in a online manner or are they pretrained by providing all available data beforehand? This should be clarified here, because it also may cause some confusion when interpreting the results in Sec. 5.3.

Section 5.3:	Fig. 3 must be revised, as the font size in the graph is too small to be readable without zooming in the PDF and is completely unreadable if printed on paper. The authors should consider only presenting one meaningful example from the plots as a full-sized graph and describing it in the text. The behavior shown in Fig. 3 is not discussed in the text, e.g., why do we see a nearly linear increase in the reward curve in some cases and a more step-wise or linear/exponential increases in others?

The results depicted in Tabs. 2 and 3 indicate that there is a huge performance difference between the MFCC and CNN features in nearly all evaluated scenarios. As the presented work focuses on the online learning aspect rather than on evaluating different features, the authors might consider only reporting the CNN-based features to make the results a little easier to grasp due to more available space.  Additionally, the baseline methods seem to yield much better performance than BerlinUCB in many situations. It is, however, unclear, if these performance differences are caused by the fact that the baselines are not trained online (see previous comment on Sec. 5.1) or if other factors are responsible for this. In general, the discussion of the results should be extended and explanations on the individual performance differences should be provided to get a better understanding of the actual performance of BerlinUCB and its self-supervised extensions. The latter should also be discussion in more detail (see previous comment on the reward curves in Fig. 3).

Lastly, the paper is missing an appropriate conclusion and outlook, which would be, for instance, a suitable place to discuss future extensions regarding branch management, improving self-supervision, different features etc.

---

### Official Review · AnonReviewer2 · 2020-06-29
**An online speaker diarization framework for a speaker recognition task is proposed that combines embedding extraction, clustering and re-segmentation into the same problem as an online decision making problem**

**Confidence:** 5
**Rating:** 5

**Review:**

Authors, in this work, proposed an online speaker diarization framework for a speaker recognition task that combines embedding extraction, clustering and re-segmentation into the same problem as an online decision making problem. Compared to the conventional clustering algorithms (Kmeans, KNN and GMM) the performances of the proposed self-supervised learning based approaches are worse in the majority of the cases.

(1) In abstract authors wrote "We proposed a novel AI framework to conduct real-time multi-speaker diarization and recognition without prior registration and pre-training in a fully online learning setting."
For the Minivox C*-CNN-12k systems, how did you generate embeddings if you did not use any pre-trained model (VGG-M)?
(2) As baseline, authors could have included i-vector/PLDA/AHC or x-vector/PLDA/AHC systems for comparison purposes.
(3) In tables 2 & 3, for Kmeans and KNN algorithms the DER increases while going from C5-*  to C10-* but again decreases for C20-*. Authors did not provide any explanation for this.
(4) As this work is about online diarization/speaker recognition, authors should provide some timing information (execution time in real-time factor) especially for processing 12k - 60k time-frames (120s - 600s assuming a frame shift of 10 ms was used)
(5) In page 2 authors wrote "A preferable AI engine for such a realistic speaker recognition and diarization system should (1) not require user registrations, (2) allow new user to be registered into the system real-time, (3) transfer voiceprint information from old users to new ones, (4) be up running without pre-training on large amount of data in advance."
For speaker recognition task, registration/enrollment is a necessary whether it is done online/offline.
(6) Authors used many acronyms without defining beforehand.

In terms of Quality of References: Misses some relevant works.
In terms of Clarity of presentation: Clear but could benefit from some revision.
In terms of Originality: Seems original to me
In terms of significance: I do not think every steps of speaker recognition/diarization has to be done online.

---

### Decision · Program_Chairs · 2020-07-01

**Decision:**

Reject

**Comment:**

Dear author(s),

Thank you very much for your submission at the ICML2020@SaS workshop (https://icml-sas.gitlab.io/). Based on the scores assigned by the reviewers, we regret to inform you that the paper was rejected. We got 26 submissions and we were only able to accept 13 papers. We invite you anyway to consider the feedback of the reviewers and to follow our upcoming workshop on July 17.

---

> ### Author Response · Authors · 2020-07-24
> **Thank you for the helpful suggestion and remarks!**
>
> We would like to thank the anonymous reviewers for the detailed and helpful suggestion, which we will take into careful consideration for the next revision of this paper. We agree with most evaluations on the brevity of many methodological aspects and advice on better interpretations for the empirical behaviors that we observed. Thank you! ~ Baihan